

# Distributions of *p*-values smaller than .05 in psychology: what is going on?

Chris H.J. Hartgerink[1], Robbie C.M. van Aert[1], Michèle B. Nuijten[1], Jelte M. Wicherts[1] and Marcel A.L.M. van Assen[1,2]

[1] Department of Methodology and Statistics, Tilburg University, Tilburg, The Netherlands
[2] Department of Sociology, Utrecht University, Utrecht, The Netherlands

## ABSTRACT

Previous studies provided mixed findings on pecularities in *p*-value distributions in psychology. This paper examined 258,050 test results across 30,710 articles from eight high impact journals to investigate the existence of a peculiar prevalence of *p*-values just below .05 (i.e., a bump) in the psychological literature, and a potential increase thereof over time. We indeed found evidence for a bump just below .05 in the distribution of exactly reported *p*-values in the journals Developmental Psychology, Journal of Applied Psychology, and Journal of Personality and Social Psychology, but the bump did not increase over the years and disappeared when using recalculated *p*-values. We found clear and direct evidence for the QRP "incorrect rounding of *p*-value" (*John, Loewenstein & Prelec, 2012*) in all psychology journals. Finally, we also investigated monotonic excess of *p*-values, an effect of certain QRPs that has been neglected in previous research, and developed two measures to detect this by modeling the distributions of statistically significant *p*-values. Using simulations and applying the two measures to the retrieved test results, we argue that, although one of the measures suggests the use of QRPs in psychology, it is difficult to draw general conclusions concerning QRPs based on modeling of *p*-value distributions.

# INTRODUCTION

A set of *p*-values can be informative of the underlying effects that are investigated, but can also be indicative of potential research biases or questionable research practices (QRPs). In the absence of QRPs, the distribution of significant *p*-values can be expected to have a certain shape. Under the null-hypothesis all *p*-values are equally probable (i.e., follow a uniform distribution). If there is truly an effect, smaller *p*-values are more likely than larger *p*-values (i.e., the distribution decreases monotonically in the *p*-value). Consequently, because some hypotheses are false and some are true, the distribution of observed *p*-values arises from a mixture of uniform and right-skewed distributions and should also decrease monotonically.[1] QRPs may have various effects on the *p*-value distribution. Figure 1 shows the *p*-value distribution of statistical tests both with data peeking (solid lines) and

Corresponding author
Chris H.J. Hartgerink,
chjh@protonmail.com

---

[1] One exception to this rule is when the alternative hypothesis is wrongly specified; that is, if the true effect size is negative whereas the alternative hypothesis states that the true effect is positive. In this case, the distribution of the *p*-value is left-skewed and monotonically increasing.

---
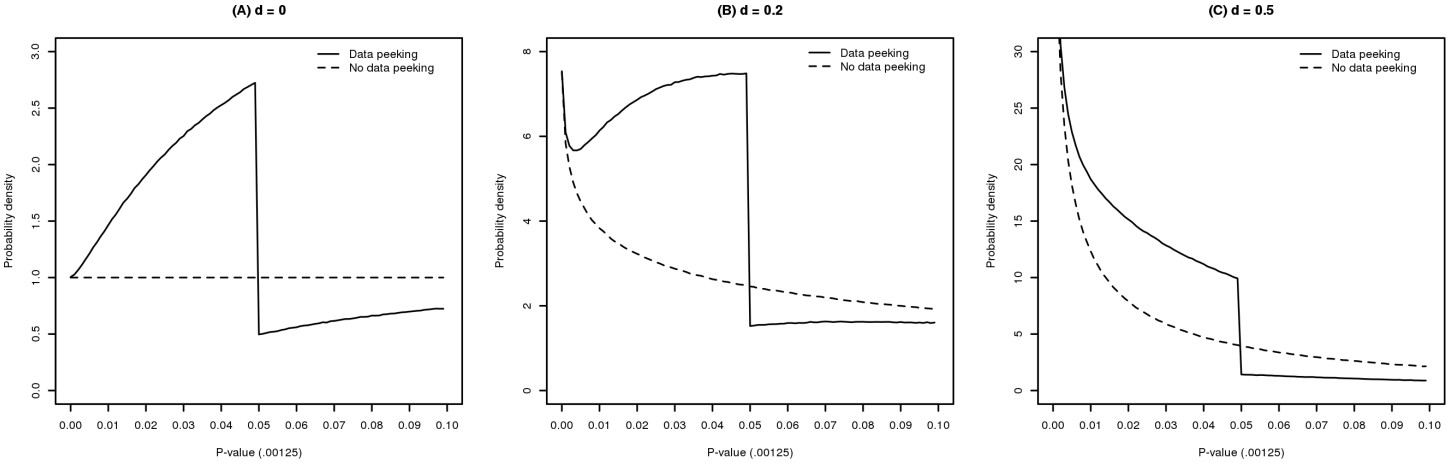

**Figure 1** **Distributions of 20 million *p*-values each, when Cohen's standardized effect size *d* = 0 (bump; A), *d* = .2 (bump; B), and *d* = .5 (monotonic excess; C), given data peeking (solid) or no data peeking (dashed).** Simulations were run for two-sample *t*-tests with $n_k = 24$. For data peeking, a maximum of three rounds of additional sampling occurred if the result was nonsignificant, with each round adding 1/3 of the original sample size.

without data peeking. Data peeking (also known as optional stopping) refers to conducting intermediate significance testing during data collection (*Armitage, McPherson & Rowe, 1969*). Data peeking greatly affects the *p*-value distribution in all panels, which can be seen from comparing the 'true' and 'data-peeked' *p*-value distributions. Figure 1A, which is obtained after data peeking of studies with standardized effect size *d* = 0, shows a 'bump' in the distribution. A bump corresponds to that part of the *p*-value distribution that makes it no longer monotonically decreasing. Figure 1B also shows a bump for data peeking of studies with *d* = 0. However, Fig. 1C shows no bump but merely monotonic excess, i.e., an increase in the frequency of *p*-values below .05 in the absence of a bump. Consequently, data peeking may either lead to monotonic excess or a bump in the distribution of *p*-values. There are other known QRPs in the analysis of data (*John, Loewenstein & Prelec, 2012*), but these have different effects on the *p*-value distribution and do not necessarily lead to a bump, as shown in Fig. 1.

In this paper we attempt to answer two questions: (1) Does a bump or monotonic excess of *p*-values below .05 exist in psychology? and (2) Did evidence for a bump increase over time in psychology? We chose to focus on psychology because of the availability of an extensive database on statistical results in psychology (used in *Nuijten et al., 2015*) and because discussions on research practices are particularly salient in this discipline (e.g., *Pashler & Wagenmakers, 2012*; *John, Loewenstein & Prelec, 2012*; *Simmons, Nelson & Simonsohn, 2011*; *Wagenmakers et al., 2012*; *Asendorpf et al., 2013*).

## How QRPs relate to distributions of *p*-values

QRPs are defined as practices that are detrimental to the research process (*Panel on Scientific Responsibility and the Conduct of Research, 1992*), with a recent focus on those which "increase the likelihood of finding support for a false hypothesis" (p. 524 *John, Loewenstein & Prelec, 2012*). Several QRPs related to significance testing are known to affect

*p*-values of statistical tests and consequently the decisions based on these tests. Specifically, particular QRPs may yield results that are just significant and can create a bump of *p*-values, such as ad hoc exclusion of outliers (*Bakker & Wicherts, 2014*), repeatedly sampling new participants and checking the results (i.e., data peeking, *Armitage, McPherson & Rowe, 1969*), including various combinations of covariates until a significant result is reached, operationalizing a measure in different ways until significance is reached (*Simmons, Nelson & Simonsohn, 2011*), or selective reporting of *p*-values (*Franco, Malhotra & Simonovits, 2015*). These QRPs have been used by many researchers at least once in their career. For instance, data peeking and the ad hoc exclusion of outliers were admitted by 63% and 38% of psychological researchers, respectively (*John, Loewenstein & Prelec, 2012*). On the other hand, other QRPs mainly yield very small and (clearly) significant *p*-values, such as analyzing multiple conditions or correlated variables and selecting only the smallest *p*-value out of this set of analyses (R Van Aert, J Wicherts & M Van Assen, 2016, unpublished data; *Ulrich & Miller, 2015*) and do not lead to a bump. To summarize, different QRPs may differently affect the distribution of statistically significant *p*-values.

However, there are at least two problems with using *p*-value distributions to examine the prevalence of QRPs. First, as we previously argued, not all QRPs lead to a bump of *p*-values just below .05. Hence, examining the distribution of *p*-values just below .05 will not inform us on the prevalence of QRPs that do not aim to obtain just significant results but yield mainly small and clearly significant *p*-values (R Van Aert, J Wicherts & M Van Assen, 2016, unpublished data; *Ulrich & Miller, 2015*). Second, the QRPs yielding just significant results do not necessarily result in a non-monotonic *p*-value distribution, that is, a distribution with a *bump*. For instance, consider Fig. 1 that shows the result of simulations done for data peeking, which is known to result in mainly just significant *p*-values (*Armitage, McPherson & Rowe, 1969*; *Lakens, 2015b*; *Wagenmakers, 2007*). Figure 1 illustrates that data peeking may result in non-monotonic excess (i.e., bump; A and B), but can also cause *monotonic excess* (C), even if all researchers use data peeking. Specifically, if all underlying effects are genuinely and substantially different from zero (C), data peeking will generally not lead to a bump below .05. In the present paper, we therefore examine the peculiar prevalence of *p*-values just below .05 by both investigating the presence of a bump or monotonic excess in distributions of statistically significant results.

## Previous findings

*Masicampo & Lalande (2012)* found a bump of *p*-values just below .05 in three main psychology journals (i.e., *Journal of Personality and Social Psychology*, JPSP; *Journal of Experimental Psychology: General*, JEPG; *Psychological Science*, PS), which, as we saw, could be explained by research biases due to QRPs. The observation of a bump was one of several signals of a crisis of confidence in research findings in psychological science (*Pashler & Wagenmakers, 2012*; *Ferguson, 2015*). *Leggett et al. (2013)* later corroborated this bump of *p*-values for JPSP and JEPG, and observed that it was larger in 2005 than in 1965. Considering that research biases can lead to overemphasis on statistical significance, this result suggested that the state of psychology may have even deteriorated over the years. Additional corroboration in samples of published articles from various fields was
provided by *Head et al. (2015)*, who documented the bump of *p*-values below .05 in 1,048,575 articles across 16 disciplines including psychology. *Ginsel et al. (2015)* found similar biased reporting of *p*-values in medical abstracts, but noted the variety of potential causes (e.g., publication bias, fraud, selective reporting).

At the same time, other studies failed to find a bump of *p*-values below .05 (*Jager & Leek, 2014*; *Krawczyk, 2015*; *Vermeulen et al., 2015*). Reanalysis of original data by *Lakens (2015b)* and ourselves indicated that the results may have been confounded by publication bias (*Masicampo & Lalande, 2012*) and tendencies to round *p*-values (*Head et al., 2015*). Publication bias refers to the fact that the probability of getting published is higher for statistically significant results than for statistically nonsignificant results (*Gerber et al., 2010*; *Franco, Malhotra & Simonovits, 2014*). Publication bias only changes the *p*-value distribution above .05 and cannot cause a bump. *Krawczyk (2015)* analyzed a sample of around 5,000 psychology articles and found no bump in *p*-values that were *recalculated* on the basis of reported test statistics and degrees of freedom (cf. *Bakker & Wicherts, 2011*). However, he did observe a bump for *reported p*-values. As such, this highlights an important difference between reported *p*-values and recalculated *p*-values, and stresses the need to distinguish both types of results when studying signs of questionable research practices.

## Extensions of previous studies

In answering our research questions, we extend previous studies on four dimensions. First, we eliminate the distortive effects of publication bias on the *p*-value distribution by inspecting only statistically significant results. Second, we use a large dataset on *p*-values from entire articles instead of only *p*-values from abstracts (as in *Jager & Leek, 2014*; *De Winter & Dodou, 2015*). Third, we distinguish between reported and recalculated *p*-value distributions for the same set of test results and show that this distinction affects answers to the two questions because of common mismatches (*Bakker & Wicherts, 2011*). Fourth, we fit analytic models to *p*-value distributions to investigate the existence of monotonic excess as shown in Fig. 1C, whereas previous research only investigated whether there was non-monotonic excess (i.e., a bump).

Publication bias distorts the *p*-value distribution, but distortions caused by this bias should not be confounded with distortions caused by other QRPs. Publication bias refers to the selective publication of disproportionate amounts of statistically significant outcomes (*Gerber et al., 2010*; *Franco, Malhotra & Simonovits, 2014*). Publication bias contributes to a higher frequency of *p*-values just below .05 relative to the frequency of *p*-values just above .05, but only does so by decreasing the frequency of *p*-values *larger* than .05. *Masicampo & Lalande (2012)* and *De Winter & Dodou (2015)* indeed found this relatively higher frequency, which is more readily explained by publication bias. QRPs that lead to a bump affect only the distribution of *p*-values smaller than .05 (*Lakens, 2015b*). We focus only on the distribution of significant *p*-values, because this distribution is directly affected by QRPs that cause a bump or monotonic excess. Publication bias only indirectly affects this distribution, through QRPs to obtain statistically significant results, but not directly because publication bias lowers the frequency of observed nonsignificant *p*-values.

The second extension is the use of more extensive data for psychology than previously used to inspect QRPs that cause a bump or monotonic excess, improving our ability to examine the prevalence of QRPs. *Masicampo & Lalande (2012)* and *Leggett et al. (2013)* manually collected *p*-values from a relatively small set of full research articles (i.e., 3,627 and 3,701), whereas *Jager & Leek (2014)* and *De Winter & Dodou (2015)* used automated extraction of *p*-values from only the abstracts of research papers. However, *p*-values from abstracts are not representative for the population of *p*-values from the entire paper (*Benjamini & Hechtlinger, 2014*; *Ioannidis, 2014*), even though some have argued against this (*Pautasso, 2010*). Our large scale inspection of full-text articles is similar to papers by *Head et al. (2015)* and *Krawczyk (2015).*

Third, we examine the prevalence of QRPs that cause a bump or monotonic excess by investigating both reported and the accompanying recalculated *p*-values. Not all previous studies distinguished results from reported *p*-values and recalculated *p*-values. This distinction is relevant, because reported *p*-values are subject to reporting bias such as rounding errors, particularly relevant around the .05 threshold. Such reporting biases result in inaccurate *p*-value distributions. For example, there is evidence that reporting errors that affect statistical significance occur in approximately 10–15% of papers in psychology (i.e., gross inconsistencies *Bakker & Wicherts, 2011*; *García-Berthou & Alcaraz, 2004*; *Nuijten et al., 2015*; *Veldkamp et al., 2014*). The advantage of analyzing recalculated *p*-values is that they contain more decimals than typically reported and that they correct reporting errors. Some previous studies analyzed reported *p*-values (*De Winter & Dodou, 2015*; *Jager & Leek, 2014*; *Head et al., 2015*), whereas others looked at recalculated *p*-values (*Masicampo & Lalande, 2012*) or a mix of reported and recalculated (*Leggett et al., 2013*). Only *Krawczyk (2015)* used both reported and recalculated *p*-values for a subset of the data (approximately 27,000 of the 135,000 were recalculated), and found that the peculiar prevalence below .05 disappeared when the recalculated data were used. Hence, this distinction between reported and recalculated *p*-values allows us to distinguish between peculiarities due to reporting errors and peculiarities due to QRPs such as data peeking.

Fourth, we examine the prevalence of *p*-values just below .05 by taking into account various models to test and explain characteristics of *p*-value distributions. We applied tests and fitted models to *p*-values below .05, in two ways. We first applied the non-parametric Caliper test (*Gerber et al., 2010*) comparing frequencies of *p*-values in an interval just below .05 to the frequency in the adjacent lower interval; a higher frequency in the interval closest to .05 is evidence for QRPs that seek to obtain just significant results. The Caliper test has also been applied to examine publication bias, by comparing just significant to just nonsignificant *p*-values (*Kühberger, Fritz & Scherndl, 2014*), and to detect QRPs (*Head et al., 2015*). However, the Caliper test can only detect a bump but not monotonic excess, as illustrated by the distributions of *p*-values in Fig. 1. Therefore, we also attempted to model the distribution of significant *p*-values in order to investigate for all forms of excess (i.e., both a bump and monotonic excess), and illustrate the results and difficulties of this approach.

In short, this paper studies the distribution of significant *p*-values in four ways. First, we verified whether a bump is present in *reported p*-values just below .05 with the Caliper test.

**Table 1** Articles downloaded, articles with extracted results in American Psychological Association (APA) style, and number of extracted APA test results per journal.

| Journal | Acronym | Timespan | Articles downloaded | Articles with extracted results (%) | APA results extracted |
|---|---|---|---|---|---|
| Developmental Psychology | DP | 1985–2013 | 3,381 | 2,607 (77%) | 37,658 |
| Frontiers in Psychology | FP | 2010–2013 | 2,126 | 702 (33%) | 10,149 |
| Journal of Applied Psychology | JAP | 1985–2013 | 2,782 | 1,638 (59%) | 15,134 |
| Journal of Consulting and Clinical Psychology | JCCP | 1985–2013 | 3,519 | 2,413 (69%) | 27,429 |
| Journal of Experimental Psychology General | JEPG | 1985–2013 | 1,184 | 821 (69%) | 18,921 |
| Journal of Personality and Social Psychology | JPSP | 1985–2013 | 5,108 | 4,346 (85%) | 101,621 |
| Public Library of Science | PLOS | 2000–2013 | 10,303 | 2,487 (24%) | 31,539 |
| Psychological Science | PS | 2003–2013 | 2,307 | 1,681 (73%) | 15,654 |
| | | *Total* | *30,710* | *16,695 (54%)* | *258,105* |

Second, to examine how reporting errors might influence $p$-value distributions around .05, we analyzed only the recalculated $p$-values corresponding to those reported as .05. Third, we used the Caliper test to examine if a bump effect is present in *recalculated* $p$-values and whether evidence for a bump changed over time. Finally, we modeled the distribution of significant recalculated $p$-values in an attempt to also detect a monotonic excess of $p$-values below .05.

## DATA AND METHODS

### Data

We investigated the $p$-value distribution of research papers in eight high impact psychology journals (also used in *Nuijten et al., 2015*). These eight journals were selected due to their high-impact across different subfields in psychology and their availability within the Tilburg University subscriptions. This selection also encompasses the journals covered by *Masicampo & Lalande (2012)* and *Leggett et al. (2013)*. A summary of the downloaded articles is included in Table 1.

For these journals, our sample included articles published from 1985 through 2013 that were available in HTML format. For the PLOS journals, HTML versions of articles were downloaded automatically with the rplos package (v0.3.8; *Chamberlain, Boettiger & Ram, 2015*). This package allows an R user to search the PLOS database as one would search for an article on the website.[2] We used this package to retrieve search results that include the subject 'psychology' for (part of) an article. For all other journals, HTML versions of articles were downloaded manually by the first author.

APA test results were extracted from the downloaded articles with the R package statcheck (v1.0.1; *Epskamp & Nuijten, 2015*). The only requirement for this package to operate is a supply of HTML (or PDF) files of the articles that are to be scanned and statcheck extracts all test results reported according to the standards of the American Psychological Association (APA; *American Psychological Association, 2010*). This format is defined as test results reported in the following order: the test statistic and degrees of

[2]We note there are minor differences in the number of search results from the PLOS webpage and the rplos package for equal searches. This is due to differences in the default search database for the webpage and the package. For technical details on this issue, see https://github.com/ropensci/rplos/issues/75.

**Table 2** Composition of extracted APA test results with respect to exact and inexact reporting of *p*-values or test statistics.

|  | Exact test statistic | Inexact test statistic |  |
|---|---|---|---|
| Exact *p*-value | 68,776 | 274 | *69,050 (27%)* |
| Inexact *p*-value | 187,617 | 1,383 | *189,000 (73%)* |
|  | *256,393 (99.36%)* | *1,657 (0.64%)* | **258,050 (100%)** |

freedom (encapsulated in parentheses) followed by the *p*-value (e.g., $t(85) = 2.86, p = .005$). This style has been prescribed by the APA since at least 1983 (*American Psychological Association, 1983*; *American Psychological Association, 2001*), with the only relevant revision being the precision of the reported *p*-value, changing from two decimal places to three decimal places in the sixth edition from 2010. statcheck extracts $t$, $F$, $\chi^2$, $Z$ and $r$ results reported in APA style. Additional details on the validity of the statcheck package can be found in *Nuijten et al. (2015)*.

From the 30,710 downloaded papers, statcheck extracted 258,105 test results. We removed 55 results, because these were impossible test results (i.e., $F(0, 55) = \cdots$ or $r > 1$). The final dataset thus included 258,050 test results. The extracted test results can have four different formats, where test results or *p*-values are reported either exactly (e.g., $p = .042$) or inexactly (e.g., $p < .05$). Table 2 shows the composition of the dataset, when split across these (in)exactly reported *p*-values and (in)exactly reported test results.

From this dataset, we selected six subsets throughout our analyses to investigate our research questions regarding a bump below .05. We analyzed (i) all reported *p*-values ($N = 258,050$) for a bump in their distribution just below .05. Subsequently we analyzed (ii) only exactly reported *p*-values ($N = 69,050$). It is possible that reporting or rounding errors have occurred among the reported *p*-values. To investigate the degree to which this happens at $p = .05$, we analyzed (iii) exactly reported test statistics that are accompanied by an exactly reported *p*-value of .05 (i.e., $p = .05$). This subset contains 2,470 results. To attenuate the effect of rounding errors and other factors influencing the reporting of *p*-values (e.g., *Ridley et al., 2007*), we also investigated the recalculated *p*-value distribution with (iv) *p*-values that were accompanied by exactly reported test statistics ($N = 256,393$). To investigate whether evidence for a bump differs for inexactly and exactly reported *p*-values, (v) 68,776 exactly reported test statistics with exactly reported *p*-values were analyzed. Finally, we used (vi) all recalculated *p*-values in 0–.05 for $t$, $r$, and $F(df_1 = 1)$ values to model the effect size distribution underlying these *p*-values to investigate evidence of both a bump and monotonic excess.

## Methods

We used the Caliper test and two new measures to examine if the observed *p*-value distribution shows evidence for a bump or monotonic excess below .05. We applied the two measures to the observed *p*-value distribution and we examined their performance to detect a bump or monotonic excess using a simulation study on data peeking. Data peeking was chosen because it is one of the most frequently used and well-known QRPs. Below,

we explain the Caliper test, how the $p$-value distributions are modeled with the two new measures, and describe the design of the simulation study in more detail.

### Caliper test

In order to test for a bump of $p$-values just below .05, we applied the Caliper test (e.g., *Gerber et al., 2010*; *Kühberger, Fritz & Scherndl, 2014*). This proportion test compares the frequencies of $p$-values in two intervals, such as the intervals .04–.045 and .045–.05. Let *Pr* denote the proportion of $p$-values of the interval .045–.05. Then, independent of the population effect sizes underlying the $p$-values, *Pr* should not be higher than .5 in any situation because the $p$-value distribution should be monotone decreasing. Hence $Pr > .5$ signifies a bump of $p$-values just below .05.

We carried out one-tailed binomial proportion tests, with $H_0 : Pr \leq .5$ and $H_1 : Pr > .5$. For example, if 40 and 60 $p$-values are observed in the intervals .04–.045 and .045–.05, respectively, then $Pr = .6$ and the binomial test results in $p$-value $= .0284$, suggesting evidence for a bump below .05. We applied the Caliper test to the reported $p$-values (subsets one through three as described in the previous section) and recalculated $p$-values (subsets four and five), both for the entire dataset and each of the eight psychology journals.

The Caliper test requires specifying the width of the intervals that are to be compared. For reported $p$-values, we selected the intervals (.03875–.04] and [.04875–.05) because there is a strong preference to report $p$-values to the second decimal in research papers (see also *Hartgerink, 2015*). For recalculated $p$-values we used the same interval width as used by *Masicampo & Lalande (2012)* and *Leggett et al. (2013)*, which is .00125, corresponding to a comparison of intervals (.0475–.04875] and [.04875–.05). Note that rounding is not a problem for recalculated $p$-values. Considering that some journals might show small frequencies of $p$-values in these intervals, we also carried out Caliper tests with interval widths of .0025, .005, and .01. Note that, on the one hand, increasing interval width increases the statistical power of the Caliper test because more $p$-values are included in the test, but on the other hand also decreases power because *Pr* is negatively related to interval width whenever $p$-values correspond to tests of non-zero population effects. In other words, a bump just below .05 will tend more and more towards a monotonically decreasing distribution as the binwidth increases.

To verify if evidence for a bump of $p$-values increased over time, we fitted a linear trend to proportion *Pr* of the Caliper test with binwidths .00125, .0025, .005, and .01. We computed these proportions for each year separately, for both the total dataset and per journal. Time was centered at the start of data collection, which was 1985 except for PLOS (2000), PS (2006; due to 0 $p$-values in the considered interval for preceding years), and FP (2010). The value .5 was subtracted from all *Pr* values, such that the intercept of the trend corresponds to the bump of $p$-values at the start of data collection, where 0 means no bump. A positive linear trend signifies an increase in the bump of $p$-values below .05 over time.

## Measures based on *p*-value distributions

Figure 1 demonstrates that the effect of data peeking on the shape of the $p$-value distribution (i.e., bump or just monotonic excess) depends on the true effect size. The distribution after

data peeking does not monotonically decrease for $d = 0$ or $d = .2$ (A and B), whereas it does decrease monotonically for $d = 0.5$ (C). Consequently, the Caliper test will signal a bump of $p$-values for $d = 0$ (i.e., it will detect a bump), but not for $d = 0.5$.

We examined how we may be able to detect both a bump and monotonic excess of $p$-values below .05. Figure 1 indicates that, for $p$-values close to zero (e.g., $\leq .00125$) the $p$-value distributions with data peeking (solid lines) closely match the $p$-value distributions without data peeking (dashed lines). In other words, data-peeking in studies with initially nonsignificant $p$-values rarely results in tiny significant $p$-values, but more often in $p$-values larger than .00125. The basic idea of this analysis is therefore to estimate the 'true' effect size distribution using only these tiny $p$-values (i.e., $\leq .00125$), assuming that none or a very small proportion of these $p$-values were affected by data-peeking. We note that we selected the .00125 cut-off point rather arbitrarily. Other, more liberal (e.g., .01, in case of a smaller set of statistically significant $p$-values) or even more conservative cut-off points (e.g., .0001, in case of a very large dataset as ours) can be selected.

We examined the performance of two measures to detect a bump or monotonic excess of $p$-values below .05. The first method compares the effect sizes estimated on $p$-values smaller than .00125 to effect sizes estimated using all $p$-values smaller than .05. The idea of this first method is that increasing the frequency of just-significant $p$-values *decreases* the effect size estimate. Indeed, the more right-skewed the $p$-value distribution, the higher the effect size estimate when keeping constant studies' sample sizes (*Simonsohn, Nelson & Simmons, 2014*; *Van Assen, Van Aert & Wicherts, 2015*). According to the first method, there is evidence suggestive of data peeking (or other QRPs leading to a bump of $p$-values just below .05) if the effect size estimate is considerably lower when based on all $p$-values than when based on only $p$-values $\leq .00125$.

The second method yields a measure of excess of $p$-values just below .05, for either a bump or monotonic excess, by comparing the observed frequency of $p$-values in the interval .00125–.05 to the predicted frequency of $p$-values in that interval. This prediction is based on the effect size estimated using the $p$-values smaller than .00125. If the ratio of observed over expected $p$-values is larger than 1, referred to as statistic $D$, then this could indicate data peeking. Statistic $D$ is calculated as

$$D = \frac{p^o_{.00125}}{1 - p^o_{.00125}} \times \frac{1 - p^e_{.00125}}{p^e_{.00125}} \tag{1}$$

with $p^o_{.00125}$ and $p^e_{.00125}$ representing the proportion of $p$-values lower than .00125 observed and expected, respectively. Note that $D$ is an odds ratio.

For both measures the expected $p$-value distribution needs to be derived and compared to the observed $p$-value distribtuion. The expected $p$-value distribution was derived by minimizing the $\chi^2$-statistic as a function of mean effect $\delta$ and standard deviation $\tau$, where it was assumed that the true effect size (Fisher-transformed correlation, $\rho_F$) is normally distributed with parameters $\delta$ and $\tau$. We only considered nonnegative values of $\delta$ because we only fitted our model to observed positive effects. See File S1 for the technical details.

*Design of simulation study*

To examine the potential of the two measures to detect data peeking, their performance was examined on simulated data with and without data peeking. We used a two-group between-subjects design with 24 participants per group ($n_k = 24$), and compared their means using a $t$-test. The performance of both measures was examined as a function of true effect size $\delta$ (0; 0.2; 0.5; 0.8) and heterogeneity $\tau$ (0; 0.15). In the data peeking conditions, data were simulated as follows: means and variances per group were simulated and a two-sample $t$-test was conducted. If this $t$-test was statistically significant (i.e., $p \leq .05$), the $p$-value was stored, otherwise the data peeking procedure was started. In this data peeking procedure, one-third of the original sample size was added to the data before conducting another two-sample $t$-test. This data peeking procedure was repeated until a statistically significant result was obtained or three rounds of additive sampling had taken place (see osf.io/x5z6u for functions used in the simulation). The simulations were stopped if 1,000,000 studies with a $p$-value below .1 were obtained for each combination of $\delta$ and $\tau$.

## RESULTS AND DISCUSSION

In this section, we report the results of our analyses in the following order for the subsets: all reported $p$-values (258,050 results), exactly reported $p$-values (69,050 results), $p$-values erroneously reported as equal to .05 (2,470 results), all recalculated $p$-values based on exactly reported test statistics (256,393 results), recalculated $p$-values based on exactly reported test statistics and exactly reported $p$-values (68,776 results), and the modeling of $p$-value distributions based on recalculated $p$-values 0–.00125 and 0–.05 (54,561 results and 127,509, respectively). These analyses apply the Caliper test to investigate evidence of a possible bump below .05. Subsequently, the results of the two measures are presented based on all recalculated $p$-values.

### Reported *p*-values

Figure 2 shows the distribution for all reported $p$-values (i.e., 258,050; white bars) and exactly reported $p$-values (i.e., 69,050; blue bars). Results of the Caliper test indicate (i) there is a bump just below .05 when considering all reported $p$-values in bins .03875–.04 versus .04875–.05, $N = 45,667, Pr = 0.905, p < .001$ and (ii) there is less evidence for a bump when considering only exactly reported $p$-values, $N = 4,900, Pr = 0.547, p < .001$. The difference in bumps between these two subsets can be explained by the amount of $p$-values that are reported as $<.05$, which is 86% of all $p$-values reported as exactly equal to .05 and 14% of all reported $p$-values.

To investigate whether this observed bump below .05 across exactly reported $p$-values originates from one or multiple journals, we performed the Caliper test on the exactly reported $p$-values per journal. Table 3 shows the results for these tests. The results indicate that there is sufficient and reliable evidence for a bump below .05 (i.e., $Pr > .5$) for the journals DP and JPSP and sufficient evidence, but debatable reliability for JAP, where the results depend on the binwidth. However, the other five journals show no evidence for a bump below .05 in exactly reported $p$-values at all. In other words, the bump below .05 in exactly reported $p$-values is mainly driven by the journals DP, JAP, and JPSP.

Hartgerink et al. (2016), *PeerJ*, DOI 10.7717/peerj.1935

**Table 3** Caliper test for exactly reported *p*-values per journal for different binwidths.

| Binwidth | 0.00125 | | | | 0.0025 | | | | 0.005 | | | | 0.01 | | | |
|---|---|---|---|---|---|---|---|---|---|---|---|---|---|---|---|---|
| | *x* | *N* | *Pr* | *p* | *x* | *N* | *Pr* | *p* | *x* | *N* | *Pr* | *p* | *x* | *N* | *Pr* | *p* |
| All | **2,682** | **4,900** | **0.547** | **<.001** | **2,881** | **5,309** | **0.543** | **<.001** | **3,308** | **6,178** | **0.535** | **<.001** | **4,218** | **8,129** | **0.519** | **< .001** |
| DP | **319** | **531** | **0.601** | **<.001** | **336** | **567** | **0.593** | **<.001** | **383** | **653** | **0.587** | **<.001** | **464** | **843** | **0.55** | **0.002** |
| FP | 96 | 193 | 0.497 | 0.557 | 105 | 227 | 0.463 | 0.884 | 141 | 304 | 0.464 | 0.906 | 215 | 458 | 0.469 | 0.912 |
| JAP | **78** | **131** | **0.595** | **0.018** | **82** | **137** | **0.599** | **0.013** | 85 | 154 | 0.552 | 0.113 | 101 | 183 | 0.552 | 0.092 |
| JCCP | 246 | 517 | 0.476 | 0.874 | 267 | 562 | 0.475 | 0.889 | 308 | 641 | 0.48 | 0.848 | 395 | 823 | 0.48 | 0.882 |
| JEPG | 147 | 285 | 0.516 | 0.318 | 159 | 310 | 0.513 | 0.346 | 195 | 375 | 0.52 | 0.235 | 258 | 509 | 0.507 | 0.395 |
| JPSP | **1,252** | **2,097** | **0.597** | **<.001** | **1,310** | **2,207** | **0.594** | **<.001** | **1,408** | **2,399** | **0.587** | **<.001** | **1,623** | **2,869** | **0.566** | **<.001** |
| PLOS | 307 | 649 | 0.473 | 0.921 | 366 | 760 | 0.482 | 0.854 | 489 | 1,000 | 0.489 | 0.766 | 744 | 1,558 | 0.478 | 0.964 |
| PS | 237 | 497 | 0.477 | 0.859 | 256 | 539 | 0.475 | 0.886 | 299 | 652 | 0.459 | 0.984 | 418 | 886 | 0.472 | 0.957 |

**Notes.**

*x*, frequency of *p*-values in .05 minus binwidth through .05; *N*, total frequency of *p*-values across both intervals in the comparison; *Pr*, $x/N$; *p*, *p*-value of the binomial test.

Significant results ($\alpha = .05$, one-tailed) indicating excess of *p*-values just below .05 and are reported in bold.

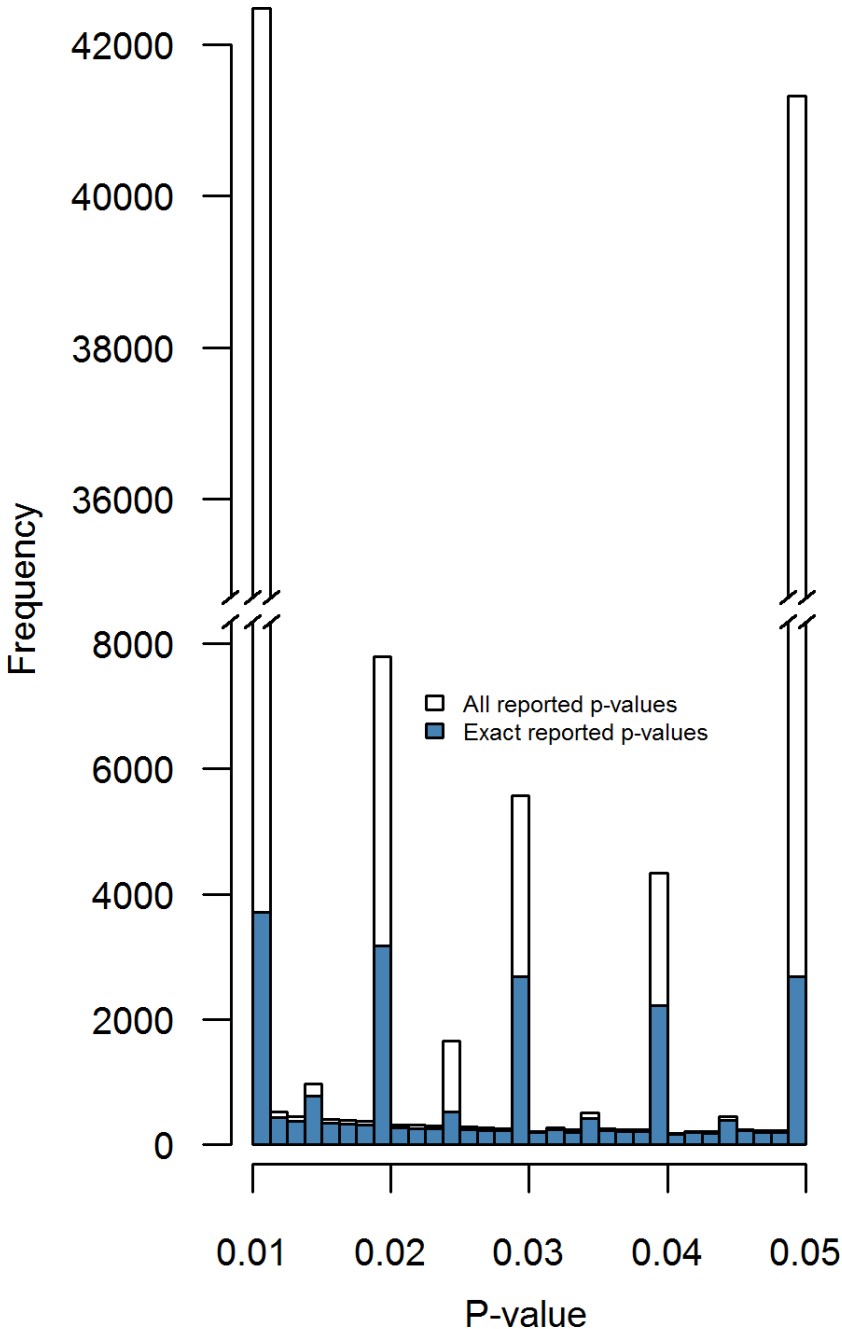

**Figure 2 Distributions of all reported *p*-values (white) and exactly reported *p*-values (blue) across eight psychology journals.** Binwidth = .00125.

The Caliper test results for reported *p*-values indicate two things: (i) including inexactly reported *p*-values has a large impact on the *p*-value distribution and (ii) a bump below .05 is also found when only considering exactly reported *p*-values. Because inexact reporting of *p*-values causes excess at certain points of the *p*-value (e.g., the significance threshold .05; *Ridley et al., 2007*), we recommend only inspecting exactly reported *p*-values when examining *p*-value distributions.

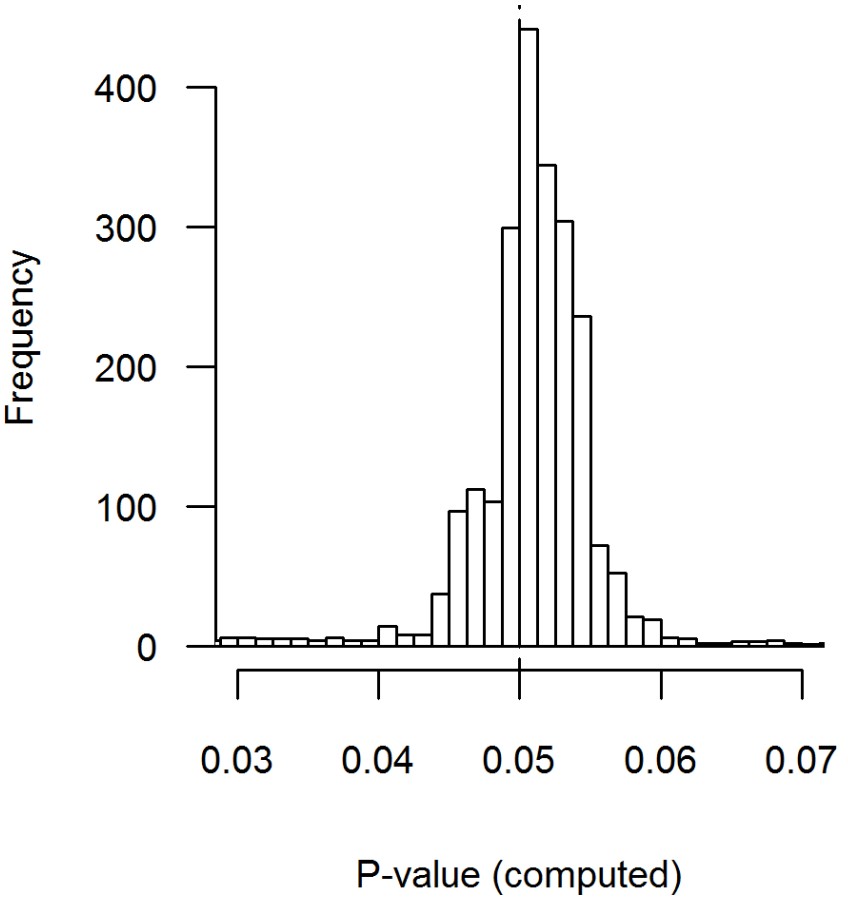

**Figure 3 Distribution of recalculated *p*-values where the *p*-value is reported as *p* = .05.** 9.7% of the results fall outside the range of the plot, with 3.6% at the left tail and 6.1% at the right tail. Binwidth = .00125

Considering only exactly reported *p*-values, there is sufficient evidence for a bump below .05 in the journals DP, JAP, and JPSP, but not in the remaining five journals (i.e., FP, JCCP, JEPG, PLOS, PS). A tentative explanation of the bump of *p*-values just below .05 for DP, JAP, and JPSP may be that QRPs that aim to obtain barely significant results are more frequent in the fields of these journals. However, another explanation may be that scientists in these fields are more prone to exactly report *p*-values just below .05 (e.g., to emphasize they are really smaller than .05) than *p*-values considerably smaller than .05.

### Recalculated *p*-value distributions
#### Recalculated when reported *p* = .05
Results for reported *p*-values remain inconclusive with regard to the distribution of *p*-values, due to potential rounding or errors (*Bakker & Wicherts, 2011*; *Nuijten et al., 2015*; *Veldkamp et al., 2014*). Rounding and errors could result in an over-representation of *p*-values ≤ .05. To investigate the plausibility of this notion, we inspected recalculated *p*-values when *p* = .05 was reported (i.e., 2,470 values). Figure 3 indicates that *p*-values that were reported as .05 show remarkable spread when recalculated, which indicates that the

reported *p*-value might frequently be rounded or incorrect, assuming that the reported test statistics are correct. More specifically, 67.45% of *p*-values reported as .05 were larger than .05 when recalculated and 32.55% were smaller than .05. This percentage does not greatly vary across journals (range 58.8%–73.4% compared to 67.45%). Taking into account rounding possibilities (i.e., widening the range of correct *p*-values to .045–.055), these percentages become 13.81% and 7.85%, respectively, meaning incorrect reporting of at least 21.66% of the *p*-values that were reported as .05. In comparison, *p*-values reported as $p = .04, p = .03$, or $p = .02$ show smaller proportions of downward rounding when compared to $p = .05$ (i.e., 53.33%, 54.32%, 50.38%, respectively compared to 67.45%). When taking into account potential rounding errors in the initial reporting of *p*-values, the discrepancy remains but becomes smaller (i.e., 11.74%, 9.57%, 8.03%, respectively compared to 13.81%). These results provide direct evidence for the QRP "incorrect rounding of *p*-value" (*John, Loewenstein & Prelec, 2012*), which contributes to a bump or monotonic excess just below .05.

The discrepancy between recalculated *p*-values and *p*-values reported as equal to .05 highlights the importance of using recalculated *p*-values when underlying effect distributions are estimated as in *p*-uniform and *p*-curve (*Van Assen, Van Aert & Wicherts, 2015*; *Simonsohn, Nelson & Simmons, 2014*). When interested in inspecting the *p*-value distribution, reported *p*-values can substantially distort the *p*-value distribution, such that results become biased if we rely solely on the reported *p*-value. Such a discrepancy indicates potential rounding of *p*-values, erroneous reporting of *p*-values, or strategic reporting of *p*-values. The *p*-value distortions can be (partially) corrected for by recalculating *p*-values based on reported test statistics. Additionally, potential distortions to the distribution at the third decimal place due to the rounding of *p*-values to the second decimal (*Hartgerink, 2015*) is also solved by recalculating *p*-values. We continue with recalculated *p*-values in our following analyses.

### Recalculated p-values

Figure 4 shows the distribution of all recalculated *p*-values (i.e., set of 256,393 results) and of recalculated *p*-values whenever the reported *p*-value is exact (i.e., set of 68,776 results). The recalculated *p*-value distribution is markedly smoother than the reported *p*-value distribution (see Fig. 2) due to the absence of rounded *p*-values.

After inspecting all recalculated *p*-values, we did not observe a bump just below .05, $N = 2,808, Pr = .5, p = 0.508$. When we analyzed the recalculated *p*-values per journal (Table 4), there is no evidence for a bump below .05 in any of the journals. Additionally, we inspected all recalculated *p*-values that resulted from exactly reported *p*-values. For this subset we did observe a bump below .05, $N = 809, Pr = 0.564, p = 0.000165$ (blue histogram in Fig. 4) for the smallest binwidth (i.e., .00125), but this effect was not robust across larger binwidths, as shown in Table 5. This table also specifies the results for a bump below .05 per journal, with sufficient evidence of a bump only in JPSP. This finding, however, was only observed for binwidths .00125 and .0025, not for larger binwidths. Considering the results from the recalculated *p*-values, there is sparse evidence for the presence of a bump below .05, opposed to previously claimed widespread evidence (*Masicampo & Lalande, 2012*;

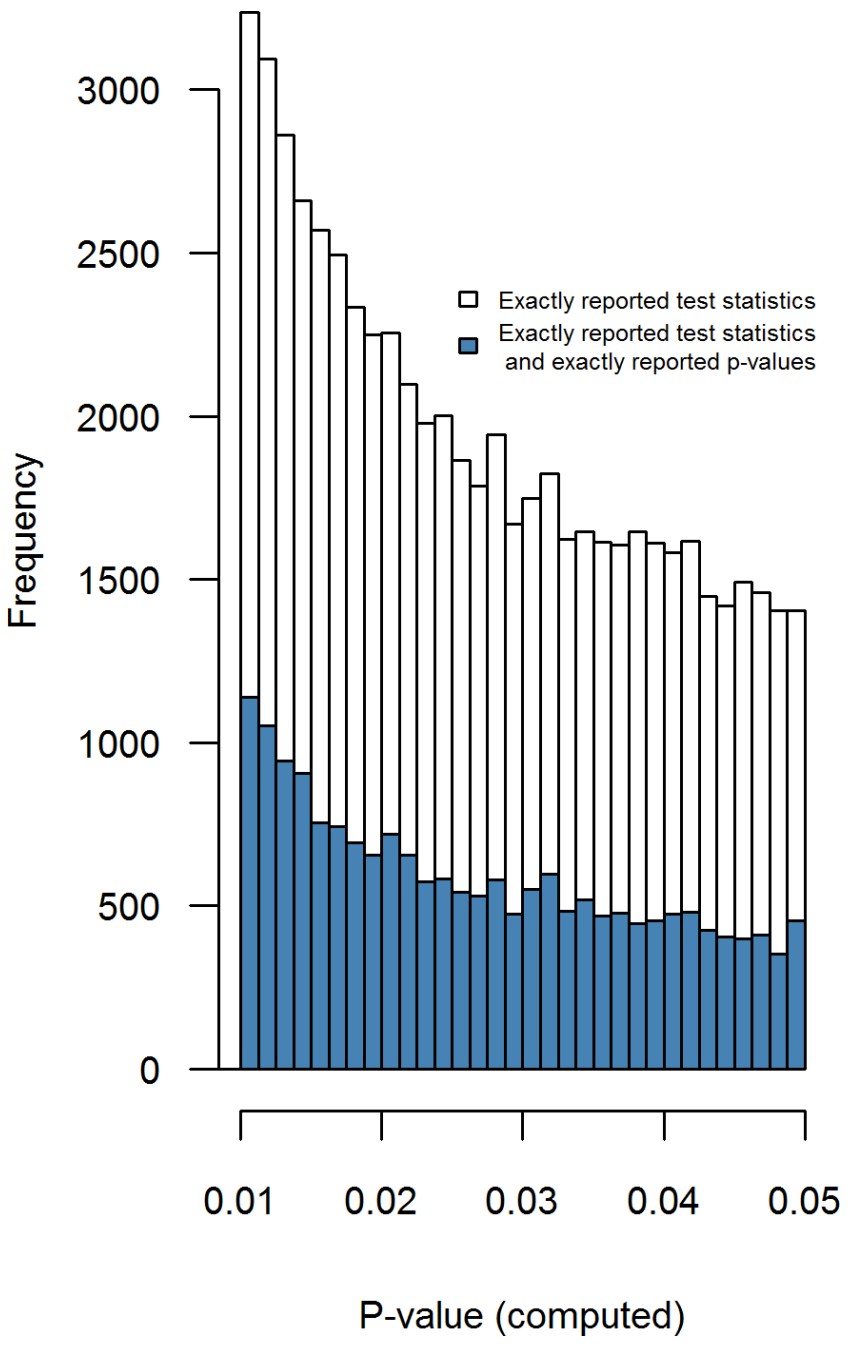

**Figure 4** Recalculated *p*-values for exactly reported test statistics (white bars), and recalculated *p*-values for exactly reported test statistics where *p*-values are also exactly reported (blue bars). Binwidth = .00125

*Leggett et al., 2013*; *Head et al., 2015*). Moreover, interpretation of the bump for JPSP is not straightforward; it may also be that authors of JPSP are more prone to report exact test statistics if the *p*-value is just below .05 than whenever *p*-values are considerably smaller than .05.

Hartgerink et al. (2016), *PeerJ*, DOI 10.7717/peerj.1935

**Table 4  Caliper test for exactly recalculated p-values per journal for different binwidths.**

| Binwidth | 0.00125 | | | | 0.0025 | | | | 0.005 | | | | 0.01 | | | |
|---|---|---|---|---|---|---|---|---|---|---|---|---|---|---|---|---|
| | *x* | *N* | *Pr* | *p* | *x* | *N* | *Pr* | *p* | *x* | *N* | *Pr* | *p* | *x* | *N* | *Pr* | *p* |
| All | 1,404 | 2,808 | 0.5 | 0.508 | 2,808 | 5,761 | 0.487 | 0.973 | 5,761 | 11,824 | 0.487 | 0.997 | 11,824 | 25,142 | 0.47 | >.999 |
| DP | 184 | 382 | 0.482 | 0.779 | 382 | 829 | 0.461 | 0.989 | 829 | 1,710 | 0.485 | 0.9 | 1,710 | 3,579 | 0.478 | 0.996 |
| FP | 30 | 69 | 0.435 | 0.886 | 69 | 172 | 0.401 | 0.996 | 172 | 376 | 0.457 | 0.956 | 376 | 799 | 0.471 | 0.955 |
| JAP | 73 | 145 | 0.503 | 0.5 | 145 | 270 | 0.537 | 0.124 | 270 | 556 | 0.486 | 0.765 | 556 | 1,168 | 0.476 | 0.952 |
| JCCP | 160 | 308 | 0.519 | 0.265 | 308 | 633 | 0.487 | 0.763 | 633 | 1,267 | 0.5 | 0.522 | 1,267 | 2,706 | 0.468 | >.999 |
| JEPG | 81 | 164 | 0.494 | 0.593 | 164 | 332 | 0.494 | 0.608 | 332 | 683 | 0.486 | 0.778 | 683 | 1,535 | 0.445 | >.999 |
| JPSP | 640 | 1,268 | 0.505 | 0.379 | 1,268 | 2,557 | 0.496 | 0.668 | 2,557 | 5,174 | 0.494 | 0.802 | 5,174 | 10,976 | 0.471 | >.999 |
| PLOS | 125 | 260 | 0.481 | 0.752 | 260 | 541 | 0.481 | 0.828 | 541 | 1,170 | 0.462 | 0.995 | 1,170 | 2,544 | 0.46 | >.999 |
| PS | 111 | 212 | 0.524 | 0.268 | 212 | 427 | 0.496 | 0.577 | 427 | 888 | 0.481 | 0.88 | 888 | 1,835 | 0.484 | 0.919 |

**Notes.**

*x*, frequency of *p*-values in .05 minus binwidth through .05; *N*, total frequency of *p*-values across both intervals in the comparison; *Pr*, *x/N*; *p*, *p*-value of the binomial test.

Significant results ($\alpha = .05$, one-tailed) indicating excess of *p*-values just below .05 and are reported in bold.

Hartgerink et al. (2016), *PeerJ*, DOI 10.7717/peerj.1935
**Table 5  Caliper tests for exactly recalculated and exactly reported *p*-values per journal, including alternative binwidths.**

| Binwidth | 0.00125 | | | | 0.0025 | | | | 0.005 | | | | 0.01 | | | |
|---|---|---|---|---|---|---|---|---|---|---|---|---|---|---|---|---|
| | *x* | *N* | *Pr* | *p* | *x* | *N* | *Pr* | *p* | *x* | *N* | *Pr* | *p* | *x* | *N* | *Pr* | *p* |
| All | **456** | **809** | **0.564** | **< .001** | 809 | 1,617 | 0.5 | 0.5 | 1,617 | 3,403 | 0.475 | 0.998 | 3,403 | 7,402 | 0.46 | 1 |
| DP | 46 | 87 | 0.529 | 0.334 | 87 | 185 | 0.47 | 0.811 | 185 | 358 | 0.517 | 0.281 | 358 | 756 | 0.474 | 0.932 |
| FP | 15 | 27 | 0.556 | 0.351 | 27 | 87 | 0.31 | > .999 | 87 | 192 | 0.453 | 0.915 | 192 | 437 | 0.439 | 0.995 |
| JAP | 8 | 20 | 0.4 | 0.868 | **20** | **29** | **0.69** | **0.031** | 29 | 65 | 0.446 | 0.839 | 65 | 141 | 0.461 | 0.844 |
| JCCP | 43 | 78 | 0.551 | 0.214 | 78 | 161 | 0.484 | 0.682 | 161 | 364 | 0.442 | 0.988 | 364 | 780 | 0.467 | 0.971 |
| JEPG | 27 | 50 | 0.54 | 0.336 | 50 | 98 | 0.51 | 0.46 | 98 | 209 | 0.469 | 0.834 | 209 | 479 | 0.436 | 0.998 |
| JPSP | **184** | **305** | **0.603** | **<.001** | **305** | **547** | **0.558** | **0.004** | 547 | 1,117 | 0.49 | 0.764 | 1,117 | 2,451 | 0.456 | > .999 |
| PLOS | 76 | 149 | 0.51 | 0.435 | 149 | 323 | 0.461 | 0.926 | 323 | 698 | 0.463 | 0.978 | 698 | 1,470 | 0.475 | 0.975 |
| PS | 57 | 93 | 0.613 | 0.019 | 93 | 187 | 0.497 | 0.558 | 187 | 400 | 0.468 | 0.912 | 400 | 888 | 0.45 | 0.999 |

**Notes.**

*x*, frequency of *p*-values in .05 minus binwidth through .05; *N*, total frequency of *p*-values across both intervals in the comparison, *Pr*, *x*/*N*; *p*, *p*-value of the binomial test.
Significant results ($\alpha = .05$, one-tailed) indicating excess of *p*-values just below .05 and are reported in bold.

**Table 6  Linear regression coefficients as a test of increasing excess of *p*-values just below .05.** Intercept indicates the degree of excess for the first year of the estimated timespan ($>0=$ excess).

|      | Timespan  | Coefficient      | Estimate   | SE        | t          | p         |
|------|-----------|------------------|------------|-----------|------------|-----------|
| All  | 1985–2013 | Intercept        | 0.007      | 0.017     | 0.392      | 0.698     |
| All  |           | Years (centered) | −0.001     | 0.001     | −0.492     | 0.627     |
| DP   | 1985–2013 | Intercept        | −0.043     | 0.056     | −0.769     | 0.448     |
| DP   |           | Years (centered) | 0.001      | 0.003     | 0.193      | 0.849     |
| FP   | 2010–2013 | Intercept        | −0.182     | 0.148     | −1.233     | 0.343     |
| FP   |           | Years (centered) | 0.055      | 0.079     | 0.694      | 0.560     |
| JAP  | 1985–2013 | Intercept        | 0.041      | 0.081     | 0.504      | 0.619     |
| JAP  |           | Years (centered) | −0.001     | 0.005     | −0.208     | 0.837     |
| JCCP | 1985–2013 | Intercept        | 0.077      | 0.058     | 1.315      | 0.200     |
| JCCP |           | Years (centered) | −0.006     | 0.004     | −1.546     | 0.134     |
| JEPG | 1985–2013 | Intercept        | −0.022     | 0.124     | −0.176     | 0.862     |
| JEPG |           | Years (centered) | 0.001      | 0.007     | 0.097      | 0.924     |
| JPSP | 1985–2013 | Intercept        | −0.002     | 0.027     | −0.062     | 0.951     |
| JPSP |           | Years (centered) | 0.000      | 0.002     | −0.005     | 0.996     |
| PLOS | 2006–2013 | Intercept        | **−0.382** | **0.114** | **−3.344** | **0.016** |
| PLOS |           | Years (centered) | **0.072**  | **0.027** | **2.632**  | **0.039** |
| PS   | 2003–2013 | Intercept        | 0.081      | 0.078     | 1.045      | 0.323     |
| PS   |           | Years (centered) | −0.009     | 0.013     | −0.669     | 0.520     |

**Notes.**
Significant results ($\alpha = .05$, two-tailed) are reported in bold.

## Excessive significance over time

The regression results of the development of a bump below .05 over time, based on recalculated *p*-values, are shown in Table 6. Results indicate that there is no evidence for a linear relation between publication year and the degree to which a bump of *p*-values below .05 is present across the different binwidths (only results for binwidth .00125 are presented; results for the other binwidths are available at http://osf.io/96kbc/). Conversely, for PLOS there is some evidence for a minor increase of a bump throughout the years ($b = .072, p = .039$), but this result is not robust for binwidths .0025, .005, and .01. These results contrast with *Leggett et al. (2013)*, who found a linear relation between time and the degree to which a bump occurred for JEPG and JPSP. Hence, based on the period 1985–2013, our findings contrast with the increase of a bump below .05 for the period 1965–2005 in psychology (*Leggett et al., 2013*). In other words, our results of the Caliper test indicate that, generally speaking, there is no evidence for an increasing prevalence of *p*-values just below .05 or of QRPs causing such a bump in psychology.

## Results of two measures based on modeling *p*-value distributions
### Simulation study
Table 7 shows the results of the two measures for data simulated with and without data peeking. The column headers show the mean effect size (i.e., $\delta$) and heterogeneity (i.e., $\tau$) of the simulated conditions, with the corresponding $\rho_F$ and $\tau_{\rho_F}$ on the Fisher transformed

**Table 7  Results of parameter estimation of the distribution of effect sizes and measures of data peeking as a function of population effect size ($\delta$, $\rho_F$), population heterogeneity ($\tau$), and data peeking, for the simulated data.** Results are based on all $p$-values 0–1, $p$-values $\leq .05$, and $\leq .00125$.

| | $p$-values | | $\tau = 0$ | | | | $\tau = .15$ | | | |
| --- | --- | --- | --- | --- | --- | --- | --- | --- | --- | --- |
| | | | $\delta = 0$<br>$\rho_F = 0$ | $\delta = .2$<br>$\rho_F = .099$ | $\delta = .5$<br>$\rho_F = .247$ | $\delta = .8$<br>$\rho_F = .390$ | $\delta = 0$<br>$\rho_F = 0$ | $\delta = .2$<br>$\rho_F = .099$ | $\delta = .5$<br>$\rho_F = .247$ | $\delta = .8$<br>$\rho_F = .390$ |
| Without data peeking | 0–1 | $\hat{\rho}_F$ | 0 | 0.103 | 0.258 | 0.413 | 0 | 0.103 | 0.258 | 0.413 |
| | | $\hat{\tau}_{\rho_F}$ | 0 | 0 | 0 | 0 | 0.077 | 0.077 | 0.077 | 0.077 |
| | 0–.05 | $\hat{\rho}_F$ | 0 | 0.103 | 0.258 | 0.413 | 0 | 0.103 | 0.258 | 0.413 |
| | | $\hat{\tau}_{\rho_F}$ | 0 | 0 | 0 | 0.001 | 0.077 | 0.077 | 0.077 | 0.077 |
| | | Misfit $\chi^2$ | 0 | 0 | 0 | 0 | 0 | 0 | 0 | 0 |
| | 0–.00125 | $\hat{\rho}_F$ | 0 | 0.103 | 0.258 | 0.413 | 0.1 | 0.107 | 0.259 | 0.413 |
| | | $\hat{\tau}_{\rho_F}$ | 0 | 0 | 0 | 0.001 | 0.025 | 0.076 | 0.077 | 0.077 |
| | | Misfit $\chi^2$ | 0 | 0 | 0 | 0 | 0 | 0 | 0 | 0 |
| | | $D$ | 1 | 1 | 1 | 1 | 1.205 | 1.006 | 1.003 | 1.001 |
| With data peeking | 0–.05 | $\hat{\rho}_F$ | 0 | 0 | 0.117 | 0.345 | 0 | 0 | 0.075 | 0.360 |
| | | $\hat{\tau}_{\rho_F}$ | 0 | 0 | 0 | 0.038 | 0 | 0.055 | 0.137 | 0.091 |
| | | Misfit $\chi^2$ | **126,267.4** | **50,298.4** | **696.6** | **101.6** | **14,867.6** | **1,209.5** | **576.3** | **340.6** |
| | | $N$ | 759,812 | 811,296 | 936,517 | 994,974 | 434,660 | 525,023 | 707,650 | 889,681 |
| | 0–.00125 | $\hat{\rho}_F$ | 0 | 0.075 | 0.218 | 0.366 | 0.066 | 0.161 | 0.283 | 0.402 |
| | | $\hat{\tau}_{\rho_F}$ | 0 | 0 | 0 | 0 | 0.036 | 0 | 0 | 0.012 |
| | | Misfit $\chi^2$ | 6.9 | 3.2 | 7.1 | **11.8** | 2 | 1.9 | 2.6 | 2.1 |
| | | $N$ | 9,729 | 21,576 | 95,615 | 350,482 | 14,791 | 34,530 | 124,991 | 366,875 |
| | | $D$ | 1.977 | 1.976 | 1.835 | 1.166 | 1.628 | 1.620 | 1.472 | 1.164 |

**Notes.**

$\hat{\rho}_F$, estimated population effect; $\hat{\tau}_{\rho_F}$, estimated population heterogeneity; misfit 0–.05; misfit of estimates based on $p$-values 0–.05, misfit 0–.00125, misfit of estimates based on $p$-values 0–.00125 (bold indicates $p < .05$); $N$, number of results included in estimation; $D$, comparison of observed- and expected $p$-value frequencies.

correlation scale. The first set of rows shows the results for the data simulated without data peeking, of which we discuss the results first.

The results for the data without data peeking inform us on (i) whether the effect size distribution parameters can accurately be recovered using only very small ($\leq .00125$) or small $p$-values ($\leq .05$), and (ii) if both measures accurately signal no data peeking. Note that $\rho_F$ is slightly overestimated due to categorizing the $p$-value distribution into 40 categories: the estimates based on all $p$-values (i.e., $\hat{\rho}_F$, first row) are slightly larger than the population parameter (i.e., $\rho_F$, column headers).

Answering the first question of accurate parameter estimates, whenever there is no heterogeneity (i.e., $\tau_{\rho_F} = 0$) both $\rho_F$ and $\tau_{\rho_F}$ are accurately recovered. When heterogeneity is non-zero, the parameters were also accurately recovered, but not when $\rho_F = 0$. Here, $\rho_F$ was overestimated (equal to .1) and $\tau_{\rho_F}$ underestimated (.025 rather than the true .077), while at the same time the misfit was negligible.

The latter result, that the effect is overestimated under heterogeneity when $\rho_F = 0$, is explained by the fact that a $p$-value distribution can accurately be modeled with an infinite range of negatively correlated values of $\rho_F$ and $\tau_{\rho_F}$. An increase in $\rho_F$ yields a more right-skewed distribution, which is hardly distinguishable from the right-skewed distribution caused by an increase in $\tau_{\rho_F}$. Hence almost identical $p$-value distributions can be generated with $(\delta, \tau)$ and some values $(\delta^*, \tau^*)$, with $\delta^* > \mu$ and at the same time $\tau^* < \tau$, or $\delta^* < \mu$ and at the same time $\tau^* > \tau$. The similar effects of both parameters on the fitted $p$-value distribution already hint at potential problems for both measures, because performance of these measures is dependent on accurate estimates of these parameters.

With respect to the second question, whether the measures accurately signal the absence of data peeking, the first measure does so in both homo- and heterogeneous conditions, whereas the second measure correctly signals absence only under homogeneity. The first measure signals data peeking if the estimate of $\rho_F$ is smaller when based on $p \leq .05$ than on $p \leq .00125$. Previously, we already noted that effect size estimates were identical to population effect sizes under homogeneity, and equal or *larger* when based on $p \leq .00125$ under heterogeneity. This suggests that the first measure behaves well if there is no data peeking (but see the conclusion section). The second measure, $D$, performed well (i.e., was equal to 1) under homogeneity, but incorrectly suggested data peeking under heterogeneity. For instance, $D = 1.205$ for $\rho_F = 0$ and $\tau = .15$, which suggests that 20.5% more $p$-values were observed in the interval .00125–.05 than were expected based on the $\hat{\rho}_F$ estimate even though no data peeking occurred. The explanation for the breakdown of the performance of $D$ is that the parameters of the effect size distribution were not accurately recovered, overestimating the average effect size and underestimating heterogeneity based on small $p$-values. This yields a lower expected frequency of higher $p$-values (between .00125 and .05), thereby falsely suggesting data peeking.

The last rows present the results obtained when data peeking does occur. First, consider the estimates of $\rho_F$ and the performance of the first measure of data peeking. The estimates of $\rho_F$ confirm that data peeking results in underestimation, particularly if the average true effect size is not large (i.e., $\delta = .2$ or .5). Moreover, downward bias of $\rho_F$ decreases when it is estimated on $p$-values $\leq .00125$ than on $\leq .05$, accurately signaling data peeking with

the first measure. For instance, if $\rho_F = .099$ and $\tau = 0$, $\hat{\rho}_F = .075$ when based on $p$-values $\leq .00125$ and $\hat{\rho}_F = 0$ when based on $p$-values $\leq .05$. Together with the good performance of this measure under no data peeking, these results suggest that the first measure may be useful to detect data keeping in practice.

Consider the estimates of $\tau_{\rho_F}$ and the performance of $D$. Similar to conditions under no data peeking, heterogeneity is grossly underestimated when using $p$-values $\leq .00125$. Hence $D$ cannot be expected to perform well under data peeking. Although $D$-values seem to correctly signal data peeking in all conditions and decrease as expected when the effect size increases, these values do not correspond to the actual values of data peeking. For instance, consider the condition with $\delta = .5$ and $\tau_{\rho_F} = .15$; of the 582,659 simulated $p$-values in interval .00125–.05, 106,241 $p$-values were obtained through data-peeking, which yields a true $D = 1.223$, which is very different from the estimated $D = 1.472$ in Table 7.

Finally, consider the (mis)fit of the estimated $p$-value distribution. Despite the considerable downward bias in heterogeneity estimate $\hat{\tau}_{\rho_F}$, the simulated $p$-value distribution is mostly well approximated by the expected $p$-value distribution, as indicated by the small values of the $\chi^2$ statistic for $p$-values in 0–.00125. Hence, good fit again does not imply accurate parameter estimates. The misfit of the estimated distribution for $p$-values $\leq .05$ is indicated by large $\chi^2$-values, particularly when the $p$-value distribution is not monotonically decreasing (which is the case for, e.g., $\delta = 0$).

To conclude, this simulation study showed that under true homogeneity both measures of data peeking can accurately signal both absence and presence of data peeking. However, under true heterogeneity, heterogeneity is underestimated and the performance of $D$ breaks down, while results suggest that comparing estimates of average effect size, the first measure, may still accurately signal both the absence and presence of data peeking.

### Applied to data of eight psychology journals

Figure 5 depicts the observed $p$-value distribution and the expected $p$-value distribution corresponding to the fitted effect size distribution based on $p$-values $\leq .00125$. Estimates for $p$-values $\leq .05$ were effect size $\hat{\rho}_F = 0$ and heterogeneity $\hat{\tau}_{\rho_F} = .183$, and $\hat{\rho}_F = .149$ and $\hat{\tau}_{\rho_F} = .106$ for $p$-values $\leq .00125$. Note that we only considered nonnegative values of $\delta$ in the estimation procedure. Misfit between observed and expected $p$-value distribution for $p \leq .00125$ was minor ($\chi^2 = 4.1$), indicating that the observed $p$-values $\leq .00125$ were well approximated by the estimated effect size distribution.

Our first measure suggests practices leading to a monotonic excess of $p$-values below .05, because the estimated effect size based on all significant $p$-values (i.e., 0) is much smaller than the supposedly more accurate estimate based on only the very small $p$-values (i.e., .183). Moreover, assuming that effect sizes are normally distributed with $\rho_F = 0$ and $\tau_{\rho_F} = .183$, combined with the degrees of freedom of the observed effects, implies that only 27.5% of all effects would be statistically significant. However, of all reported $p$-values, 74.7% were statistically significant, but this difference may at least partly be caused by other factors such as publication bias. It is highly unlikely that the average true effect size underlying statistically significant results in psychology is truly zero. It remains undecided, however, whether this very low estimate is mainly due to QRPs leading to a downward bias

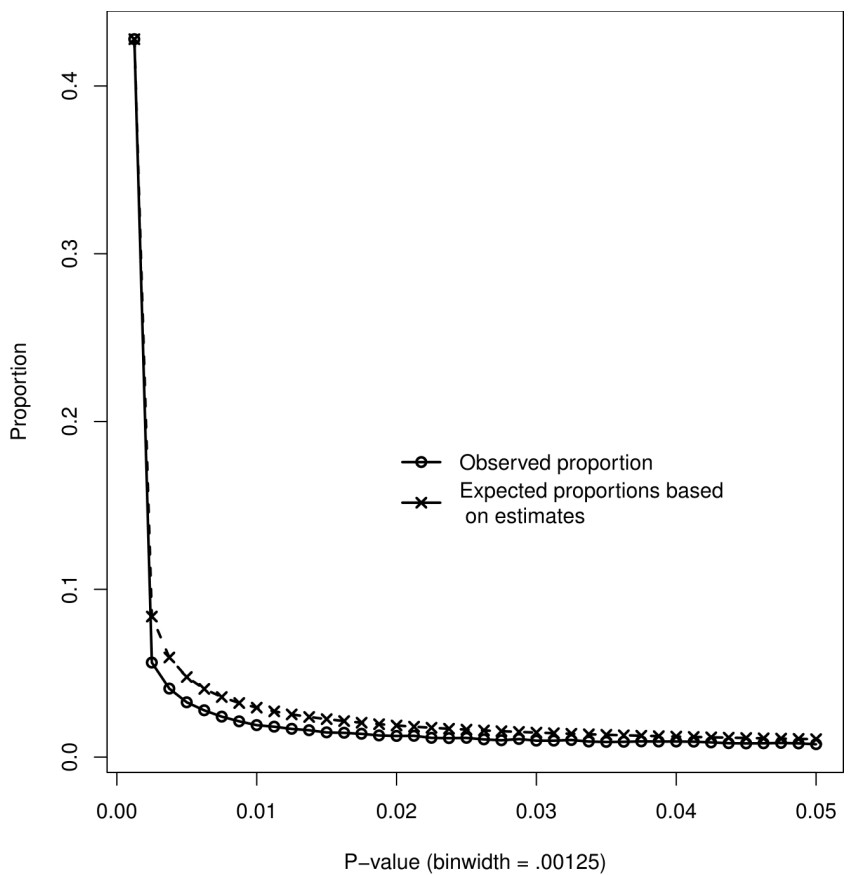

**Figure 5** Observed proportions of *p*-values (circles) and expected proportions of *p*-values based on $\hat{\rho}_F$ and $\hat{\tau}_{\rho_F}$ estimated from 0–.00125 (crosses).

of the effect size estimate, or to a misspecification of the model, an issue we revisit later in the paper.

For the second measure that compares the ratio of observed and expected *p*-values below .05, we found $D = .701$, which does not suggest data peeking but *under*-reporting of *p*-values (29.9%) in the *p*-value interval .00125–.05. The simulation results, however, have already demonstrated that the measure $D$ performs badly under effect size heterogeneity. Since heterogeneity is underlying the observed data, we conclude that the measure $D$ is not useful for investigating evidence of a bump or monotonic excess of *p*-values.

## LIMITATIONS AND CONCLUSION

Before concluding, some limitations of our method to collect *p*-values need to be addressed. First, `statcheck` (*Epskamp & Nuijten, 2015*; *Nuijten et al., 2015*), the R package used to collect the observed data, extracts all APA test results reported in the text of an article, but not those reported in tables. Hence, our selection of results is potentially not representative of all reported results and systematically excludes results that are not reported to APA standards. Second, our analysis assumed that test statistics other than *p*-values were accurately reported. If test statistics and degrees of freedom are incorrectly reported,
recalculated $p$-values are wrong as well. We identified some erroneous test statistics (e.g., $df_1 = 0$ and $r > 1$), but do not know how often errors in reported test statistics and $df$ occur and how these errors may have affected our results. We assumed that $p$-value errors were made due to the overemphasis on them in current day research.

In light of conflicting findings and interpretations, we aimed to provide final answers to the questions (1). Does a bump or monotonic excess of $p$-values below .05 exist in psychology? and (2) Did evidence for a bump increase over time in psychology? Answering these research questions may inform us on the prevalence of QRPs and its development over time in psychology. Using statcheck, we extracted and analyzed 258,050 test results conforming to APA-style across 30,710 articles from eight high impact journals in psychology, and distinguished between results with inexactly reported $p$-values, exactly reported $p$-values, and recalculated $p$-values. The basic idea underlying our analyses is that QRPs distort the $p$-value distribution. We argued that only some QRPs yield an excess of $p$-values just below .05, and show that QRPs sometimes yield a bump and sometimes only monotonic excess of $p$-values just below .05. We used the Caliper test to test for a bump, and suggested two measures to examine monotonic excess.

Starting with the existence of a bump in psychology, we drew the following conclusions. First, *inexactly* reported $p$-values are not useful for analyses of $p$-value distributions. Second, a bump in *exactly* reported $p$-values indeed exists in psychology journals DP, JAP, and JPSP. QRPs leading to just significant $p$-values can explain these bumps, but we also cannot rule out the explanation that scientists in these particular journals are more prone to exactly report $p$-values just below .05 (e.g., to emphasize they are really smaller than .05) than $p$-values considerably smaller than .05. Third, contradicting *Leggett et al. (2013)*, the bump and evidence of a bump in psychology did not increase over the years. Fourth, when analyzing only the *exactly* reported $p$-values equal to .05, clear and direct evidence was obtained for the QRP ''incorrect rounding of $p$-value'' (*John, Loewenstein & Prelec, 2012*). Evidence of this QRP, which contributed to the bump in exactly reported $p$-values in psychology, was found in all psychology journals. Fifth, after removing reporting errors and analyzing the *recalculated* reported $p$-values, evidence of a bump was found only for JPSP. Again, this may have been caused by QRPs or by scientists being more prone to report all test statistics when $p$-values are just below .05 than if they are considerable smaller than zero.

The conclusions obtained with the two measures investigating the bump and monotonic excess are not satisfactory. First, performance of both measures is dependent on accurately recovering parameters of the effect size distribution, which turned out to be difficult; estimates of effect size heterogeneity and average effect size are highly correlated and unstable when based on only statistically significant findings. Second, simulations show that one of the measures, $D$, does not accurately assess the QRP data peeking when effect sizes are heterogeneous. Third, even though performance of the second measure (i.e., difference between effect sizes based on contaminated and supposedly uncontaminated $p$-values) is affected by estimation problems, it correctly signaled data peeking in the simulations. Fourth, when applying the second measure to the observed distribution of significant $p$-values in psychology, the measure found evidence of monotonic excess of $p$-values; the

average effect size estimate based on all these *p*-values was 0, which seems very unrealistic, and suggests the use of QRPs in psychology leading to *p*-values just below .05.

Notwithstanding the outcome of the second measure, suggesting QRPs that cause monotonic excess, we do not consider it as direct evidence of such QRPs in psychology. Lakens (p.3; 2015) suggests that "it is essential to use a model of *p*-value distributions before drawing conclusions about the underlying reasons for specific distributions of *p*-values extracted from the scientific literature." We explicitly modeled the effect size distribution and by using the degrees of freedom of test results also model the effect sizes' power and the *p*-value distribution. But we fear this is not and cannot be sufficient. First of all, we could not accurately recover the effect size distribution under heterogeneity in our simulation study, even if all assumptions of our model were met. This rendered measure *D* unfruitful when there is heterogeneity, and severely limits the usefulness of the second measure that compares estimated average effect sizes. Second, devising other models may yield other results and thereby other interpretations (*Benjamini & Hechtlinger, 2014*; *Goodman, 2014*; *Lakens, 2015a*; *De Winter & Dodou, 2015*).

Results of all the aforementioned models are most likely not robust to violations of their assumptions. For instance, we assume a normal distribution of true effect sizes. This assumption is surely violated, since the reported *p*-values arise from a mixture of many different types of effects, such as very large effects (manipulation checks), effects corresponding to main hypotheses, and zero effects ('control' variables). Additionally, consider the QRPs themselves; we examined the effect of only one QRP, data peeking, in one of its limited variants. Other QRPs exist that also increase the prevalence of *p*-values just below .05, such as multiple operationalizations of a measure and selecting the first one to be significant. Other QRPs even increase the frequency of very small *p*-values (R Van Aert, J Wicherts & M Van Assen, 2016, unpublished data). We deem it impossible to accurately model QRPs and their effects, considering the difficulties we already demonstrated for modeling the *p*-value distribution generated using a single QRP that was clearly defined. To conclude, we fear that *Gelman & O'Rourke (2014)* may be right when suggesting that drawing conclusions with regard to any QRP based on modeling *p*-value distributions obtained from automatically extracted results is unfruitful.

On the other hand, we do recommend modeling effect size and *p*-value distributions of results that all intend to test the same hypothesis, to prevent contamination by irrelevant test results (*Bishop & Thompson, 2016*; *Simonsohn, Simmons & Nelson, 2015*). Examples of methods that focus on similar results are *p*-uniform (*Van Assen, Van Aert & Wicherts, 2015*) and *p*-curve (*Simonsohn, Nelson & Simmons, 2014*), which model statistically significant statistics pertaining to one specific effect and estimate the effect size based on these statistics while correcting for publication bias. Further research should reveal if both methods can also be used to detect and correct for *p*-hacking in the context of estimating one particular effect size. Preliminary results suggest, however, that detection and correcting for *p*-hacking based on statistics alone is rather challenging (R Van Aert, J Wicherts & M Van Assen, 2016, unpublished data).

### Funding

The preparation of this article was supported by Grants 406-13-050 (Robbie C.M. van Aert) and 016-125-385 (Jelte M. Wicherts) from the Netherlands Organization for Scientific Research (NWO). The funders had no role in study design, data collection and analysis, decision to publish, or preparation of the manuscript.

### Grant Disclosures

The following grant information was disclosed by the authors:
Netherlands Organization for Scientific Research (NWO): 406-13-050, 016-125-385.

### Competing Interests

The authors declare there are no competing interests.

### Author Contributions

- Chris H.J. Hartgerink and Robbie C.M. van Aert conceived and designed the experiments, performed the experiments, analyzed the data, contributed reagents/materials/analysis tools, wrote the paper, prepared figures and/or tables.
- Michèle B. Nuijten analyzed the data, reviewed drafts of the paper.
- Jelte M. Wicherts reviewed drafts of the paper.
- Marcel A.L.M. van Assen conceived and designed the experiments, analyzed the data, contributed reagents/materials/analysis tools, wrote the paper.

### Data Availability

Open Science Framework (OSF), all analysis code available here (runs all the analyses in the paper): https://osf.io/pvrtx/. Data is directly imported from the *Nuijten et al. (2015)* paper in the analysis code (data itself is available from: https://osf.io/gdr4q).

### Supplemental Information

Supplemental information for this article can be found online at http://dx.doi.org/10.7717/peerj.1935#supplemental-information.

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
