# Peer review of "Distributions of p-values smaller than .05 in psychology: what is going on?"

_PeerJ, doi:10.7717/peerj.1935_

## Round 0.1 · original submission · Major Revisions

As pointed out by the reviewers the topic of this paper is interesting and the paper is well written. However, the paper could benefit from some changes as detailed below.

Reviewer 1 ·

Basic reporting

The manuscript is well written but could benefit from a bit of strong editing, mostly for brevity and clarity. In particular, the introduction could be reorganized to make the authors’ argument stronger, more direct, and easier to follow. Specific suggestions for reorganization and clarification are at the bottom of this section.

Figure 1 was very helpful, but it seems that the Introduction would benefit from a schematic illustrating the difference between a “bump” and a “monotonic increase”, perhaps for more than two effect sizes. I found the difference between a “bump” and a “monotonic increase” difficult to visualize at first, and this seems to be a crucial concept for the entire piece. Perhaps another figure could be added rather than trying to make Figure 1 play too many roles and to make sure the reader fully understands the ideas before going forward. It seems that Table 1 could appear in supplementary material rather than the body of the manuscript. The contingency table in Table 2 is interesting, but I’m not sure that it warrants the space taken up as a table. It doesn’t seem like the dependence between exact p-value reporting and exact test statistic reporting is crucial to the authors’ conclusions.

Suggestions:
This is a big topic, and choosing the best method to begin the introduction isn’t an easy decision. Yet I think some context is very important here. In line 11, “a set of p-values” isn’t very descriptive. Are we talking about observed p-values? Published p-values? Reported p-values? I also think it would be helpful to introduce to the reader right up front to the idea of the shape of the distribution of reported/published p-values. For example, although the technical details could be elaborated upon later, the information in the paragraph starting on line 65 would be very helpful to let the reader know that (1) it’s impossible to know exactly what the true/expected p-value distribution looks like, but (2) we know that it will always decrease monotonically. Then the reader could be informed that some studies have found evidence that the distribution of published p-values in some journals/fields/whatever don’t follow this, and that some of them have used this evidence to infer the presence of QRPs. (I’m not sure that we need the history in as great detail as is provided. I’m more concerned with the logical train of thought.) I think it’s also important to be clear about what a “bump” is, and what a “monotonic excess” is. This is where schematics would be useful. Additionally, although the introduction is full of good information, it would be helpful to streamline it and include only the information relevant to this particular study (for example, only data peeking instead of all QRPs, perhaps).

Experimental design

I found the research questions confusing, both in how they were stated and how it appears they were intended. Are the authors looking for either a bump, a monotonic excess, or both, or neither? This should be clarified. And are the authors looking for increasingly strong evidence for a bump over time? Or evidence that the bump itself is increasing? And if the latter, what does it mean for a bump to increase? Does it get narrower? Wider? Taller?

I found the evidence presented in Figure 1 to be incomplete. At what value of d does the distribution of significant p-values change from a bump to a monotonic excess in the presence of peeking?

The description of the Caliper test is clear and relevant.

For the section “measures based on p-value distributions,” I think readers will need a bit more convincing before they agree with the line of arguments. It seems that the authors are choosing an empirical cutoff of p = .00125 based on eyeballing simulations for a hypothetical situation with a fixed sample size, effect size, and data-peeking rule. Next, it’s not clear to me why a data-peeking practice would decrease my effect size estimate compared to data analysis practices where I did not data peek. The second method is equally confusing, I’m afraid. It’s not clear how tiny p-values can yield estimated effect sizes that then predict the frequency of the p-values in the entire interval. The logic here may be sound, but it needs to be fleshed out and supported with more evidence/derivations/mathematics/etc.

For the section “Modeling p-value distributions,” I again believe that readers will need more convincing. Equation 5 is particularly opaque, and it’s not clear how the sampling distribution of Fisher’s transformed Z can be used to model the expected p-value distribution. It seems that the sampling distribution assumes a fixed population with a fixed rho/effect size. But empirically, the distribution of observed effect sizes publishes in psychology journals is far from homogenous. Again, the logic may be sound and even apparent to some researchers, but if nothing else it would benefit from connecting the dots very clearly to show readers the argument.

Validity of the findings

The results for the Caliber test under the section “reported p-values” could benefit from more focus. It’s not immediately clear how the conclusions in the paragraph in lines 299 follow from the results. Table 3 seems excessively detailed and could perhaps be summarized or moved to supplemental material.

The section for “excessive significance over time” suffers from an excess of detail. I think the manuscript would benefit from cutting this research question entirely and saving it for a future manuscript.

The results for the section “results of two measures based on modeling p-value distributions” puzzled me. The p-values were used to estimate one effect size for all the analyses in the database which yielded a significant p-value? And this estimated effect size was 0? (Or is this the estimated mean effect size? Either way, why is it reasonable that this would be exactly 0?) The conclusions in the paragraph beginning on line 376 are also confusing. All of this seems to be extremely empirical, which is fine – but measures derived empirically should be first well validated. The results of the simulation study should be presented earlier so that the reader trusts the measures and understands their limitations.

The results of the “Simulation study” also seemed to suffer from a surfeit of detail and lack of coherence. The two measures need to be clearly supported by simulation studies and/or other lines of support before they are used to draw conclusions from actual data.

The conclusions should also be streamlined, and main points highlighted.

·

Basic reporting

Overall, the paper is very well written. Several minor remarks follow.

Figure 1: I believe the captions should be self-contained, so maybe the authors should remind what d stands for (especially given that they refer to figure 1 before explaining this in the main text). Is the solid line in the right-hand panel to the right of .05 perfectly flat? Should it not be sloping downwards (and convex)?

Lines 92-100. If a paragraphs opens with “we improve previous studies on … dimensions” or a similar statement, the reader will expect that the present study indeed beats all (or almost all) previous ones on these dimensions. Perhaps the authors should emphasize that (unless I am missing something) actually only some previous papers are inferior on the dimensions they mention (specifically dimensions two and three). The same applies to “Our large scale inspection of full-text articles is similar to papers by Head et al. (2015) and Krawczyk (2015), but with the addition of being able to recalculate all p-values from the test statistics extracted from the text” (lines 118-120) – the latter also did this, as the authors note elsewhere.

L 101-111. I am not sure if contrasting QRPs and publication bias is the best way to frame the problem. Many QRPs are probably driven mostly by publication bias, so it would appear more justified to speak of direct effect of publication bias vs. QRPs or abandon this comparison altogether.

L 118-122 – the authors first briefly mention the recalculated values and then “introduce” them as the main topic of the new paragraph, as if this has not been mentioned before.

L 132-133 Krawczyk (2015)… they (?)

Table 1: the reader may wonder what “APA results” exactly are

L 190 “debilitate” (?)

L 217-220 – the authors seem to provide the same information twice.

l 238-239, l 289-291 these sentences seem unclear

l. 460 “We identified some erroneous test statistics (e.g., d f1 = 0 and r > 1), but do not know how often these errors occur and how they may have affected our results.” – this seems misleading. I guess the authors meant that we do not know how often errors occur that do not lead to inadmissible values such as r>1.

Experimental design

L 157-158 "These eight journals were selected due to their high-impact across different
158 subfields in psychology and their availability within the Tilburg University subscriptions." -- this is somewhat unconvincing, especially in view of the fact that a) this research field is not in its infancy any more, so a more comprehensive approach to sample selection that could shed more light on the mechanism behind "bumps" would be expected and b) the authors seem to have developed an extraction method that has relatively low marginal search costs.

I also have some doubts about the method used to develop the "theoretical" distribution of p-values. It's just based on too many assumptions that are not likely to be satisfied.

Validity of the findings

The authors should be praised for their candid discussion of the limitations of the study. The only addition that could be considered in this respect is that the authors comment (in the first paragraph of "LIMATIONS AND CONCLUSION" perhaps) on the fact that they only harvest statistical results reported using APA style. In practice, these recommendations are often violated, at least in some journals, as documented in previous studies. This is probably not a random selection, which might have some bearing on the results.

Additional comments

Table 3: I was wondering if the findings for specific journals could be linked to their publication policies. E.g. “PLOS ONE editorial decisions do not rely on perceived significance or impact, so authors should avoid overstating their conclusions.”

Reviewer 3 ·

Basic reporting

The authors presentation is generally appropriate with sufficient background and the simulation study to display a hypothetical p-value distribution under differing selection bias schemes (Figure 1).

I think the presentation of the p-value distribution estimation may be too technical for most of the audience. Potentially this could be moved to an appendix.

Experimental design

While in the background, the authors describe may other ways that adaptive p-values can be lead to low-end bias in the distribution of p-values (multiple testing, multiple transformations, removing bad observations), I think a major issue that would interesting to look at would be the multiple comparison selection bias. This would be interesting given the data collection technique was based on text representation of p-values, not in tables. For instance, often the most significant results of tables are discussed.

Validity of the findings

Did the authors assess a sample of articles what fraction of the p-values (even in text) were missed by the automated procedure due to non APA standard inclusion. Did this vary by journal type? For fun, what p-values from this manuscript would have been selected by the automated procedure?

Potentially the table based p-values could be considered at least in some secondary analysis on some subset of papers. They could be informative in the nature of QRPs and reasons for monotone increase of low p-values.

I am may have missed it, but how were the multiple p-values within a given manuscript addressed in binwidth p-value calculation (say table 3). Presumably, there may be some correlation within manuscript?

Additional comments

No Comment.

---

## Round 0.2 · accepted · Accept

We consider that the answers for the reviewers' comments were adequate and the paper was modified accordingly.